# Advanced Strategies for Therapeutic Targeting of Wild-Type and Mutant p53 in Cancer

**DOI:** 10.3390/biom12040548

**Published:** 2022-04-06

**Authors:** Shengliang Zhang, Lindsey Carlsen, Liz Hernandez Borrero, Attila A. Seyhan, Xiaobing Tian, Wafik S. El-Deiry

**Affiliations:** 1Laboratory of Translational Oncology and Experimental Cancer Therapeutics, Warren Alpert Medical School, Brown University, Providence, RI 02912, USA; lindsey_carlsen@brown.edu (L.C.); liz_hernandez_borrero@alumni.brown.edu (L.H.B.); attila_seyhan@brown.edu (A.A.S.); xiaobing_tian@brown.edu (X.T.); 2Department of Pathology and Laboratory Medicine, Warren Alpert Medical School, Brown University, Providence, RI 02912, USA; 3Joint Program in Cancer Biology, Lifespan Health System and Brown University, Providence, RI 02912, USA; 4Legorreta Cancer Center, Brown University, Providence, RI 02912, USA; 5Pathobiology Graduate Program, Warren Alpert Medical School, Brown University, Providence, RI 02912, USA; 6Hematology-Oncology Division, Department of Medicine, Lifespan Health System and Brown University, Providence, RI 02912, USA

**Keywords:** P53, mutant p53, targeting therapy, immunotherapy, cancer

## Abstract

*TP53* is a tumor suppressor gene that encodes a sequence-specific DNA-binding transcription factor activated by stressful stimuli; it upregulates target genes involved in growth suppression, cell death, DNA repair, metabolism, among others. *TP53* is the most frequently mutated gene in tumors, with mutations not only leading to loss-of-function (LOF), but also gain-of-function (GOF) that promotes tumor progression, and metastasis. The tumor-specific status of mutant p53 protein has suggested it is a promising target for cancer therapy. We summarize the current progress of targeting wild-type and mutant p53 for cancer therapy through biotherapeutic and biopharmaceutical methods for (1) boosting p53 activity in cancer, (2) p53-dependent and p53-independent strategies for targeting p53 pathway functional restoration in p53-mutated cancer, (3) targeting p53 in immunotherapy, and (4) combination therapies targeting p53, p53 checkpoints, or mutant p53 for cancer therapy.

## 1. Introduction

The p53 protein is a sequence-specific DNA-binding transcription factor regulating gene expression related to multiple cellular functions including, but not limited to, cell cycle arrest, cell apoptosis, cell growth, DNA repair, cell metabolism, and the immune response in response to stressful stimuli [1]. Clinical studies show that wild-type p53 sensitizes cancer cells to conventional chemoradiotherapy, for example, in the treatment of rectal cancer. Nearly half of tumors harbor mutant p53. Most *TP53* mutations are single missense mutations in its functional domains, most commonly the central DNA-binding domain. Missense mutations cause mutant p53 protein to lose its wild-type functions (LOF) and acquire dominant-negative activities and endow mutant p53 with gain-of-function (GOF) capabilities that confer aggressive tumor behavior and drug resistance. Clinical studies correlate poor prognosis of cancer patients with mutant p53. p53 is considered as a biomarker for tumor progression and an excellent target for designing cancer therapeutic strategies. Even in cancers carrying wild-type p53, aberrant p53 pathway signaling often occurs due to abnormal regulation such as due to high MDM2 expression often from gene amplification. The different p53 status between cancer cells and normal cells has led to p53 becoming one of the most important and rational targets for cancer therapy. 

Over the past two decades, studies have revealed p53 structure, function, and role in tumorigenesis and development [1,2,3]. These fundamental studies strongly support the design and development of new approaches for targeting p53 (mutant and wild-type) in cancer therapy [4,5,6]. Delivery of wild-type p53 by adenovirus infection kills cancer cells and suppresses tumor growth in pre-clinical and clinical studies [7]. Peptides targeting p53 can rescue p53 function and reduce tumor proliferation [8]. Reactivation of p53 by blocking p53-MDM2/MDMX(MDM4) interaction is one of the most promising approaches for cancer therapy, and MDM2 inhibitors (such as AMG232 or milademetan) and the dual inhibitors of MDM2/MDMX (such as ALRN6924) are currently undergoing clinical evaluation [9,10]. APR-246 and COTI-2 are small molecules targeting mutant p53 under clinical evaluation in different clinical trials [11,12,13,14]. Immunotherapy targeting mutant p53 to suppress tumor growth is becoming a promising approach in cancer therapy. These pioneering studies shed light on prospective cancer therapy that targets mutant and wild-type p53. 

## 2. p53 Structure and Mutations Exploited for Drug Development

The frequency of *TP53* mutations is approximately 50% across all cancers, however, this frequency varies greatly depending on the cancer type. *TP53* mutations are most common in ovarian (47.8%), colorectal (43.2%), esophageal (43.1%), head & neck (40.6%), and laryngeal (40.4%) tumors, while cervix (5.8%), hematopoietic (12.7%), and endocrine gland (14.6%) tumors experience the lowest frequency of p53 mutations [15]. High-grade serous ovarian cancer has a p53 mutation rate approaching 95%. 

*TP53* mutations generally occur in the DNA-binding domain, with about 30% arising in specific hotspots including R175, G245, R248, R249, R273, and R282 (individual mutation rates range from ~3–7%) [16]. Clinical studies demonstrate that in many tumor types, the presence of mutated p53 correlates with a worse patient prognosis as compared to the presence of wild-type p53. Though this is still a topic of controversy in some contexts, it is most widely accepted that tumors with mutated p53 predict worse clinical outcomes in breast, head and neck, liver, hematopoietic, and lymphoid system cancers as compared to their wild-type counterparts [17]. One factor contributing to the complexity of this topic is that *TP53* mutant or variant sequences can have unique effects on cellular function. For example, cancer-associated polymorphisms in codon 72 of *TP53* can result in a non-conservative amino acid change to either proline (p53pro) or arginine (p53arg), which have different characteristics. p53pro is less efficient in inducing apoptosis compared to p53arg, while p53arg is more likely to be degraded by the human papillomavirus (HPV) E6 protein. Much work by Maureen Murphy has uncovered racial, ethnic, and geographic differences in the *TP53* codon 72 polymorphism and others. The cancer-associated mutations in *TP53* are most often missense mutations in the central DNA-binding domain. These mutations also vary greatly in terms of their impact on p53 function. Some mutations cause loss of p53 function, some provide p53 the ability to inactivate wild-type p53 that is expressed from the remaining wild-type allele (dominant-negative p53 mutations), and still others cause p53 to acquire oncogenic GOF properties [18]. These unique characteristics of different p53 mutants make mutant p53 a challenging drug target and also necessitate consideration of individual mutations when using mutant p53 as a prognostic biomarker [17]. 

### 2.1. p53 Structure Exploited for Drugs Reactivating p53 Signaling

Wild-type p53 consists of a p53 transactivation domain at its N-terminal region and a tetramerization domain at its C-terminal region (Figure 1). The N- and C-terminals flank the DNA binding domain, also called the core domain [3]. 

DNA-binding domain (amino acid residues 94–292) of the p53 protein is responsible for p53 binding in a DNA sequence-specific manner to gene regulatory regions to control gene expression (Figure 1). Functionally significant upregulated target genes include p21 (WAF1), which regulates the cell cycle; Bax, DR5, Puma, and Noxa, which regulate cellular apoptosis in response to cellular stresses; TIGAR, which regulates processes related to cellular metabolism; and IRF5/9, which regulates processes related to intracellular immune functioning [1]. p53 binds specifically to the consensus DNA sequence (5′-RRRCWWGYYY-(N = 0–13)-RRRCWWGYYY-3′) [19]. This discovery has been widely applied to engineer different vectors expressing reporter genes under the control of p53 response elements (p53RE), in addition to predicting endogenous p53 DNA-binding response elements. The use of p53 transcriptional activity reporters has provided a useful tool in the high-throughput screening (HTS) of p53-targeting drugs via functional cell-based assays. A tool was developed to identify a series of small molecules that reactivate the downstream p53 pathway in tumors with mutated p53 [20]. Multiple small molecular weight chemical compounds targeting p53 (wild-type and mutant) have been identified using this method such as NSC59984 [21], prodigiosin [22], and CB002 [23,24]. 

The p53 tetramerization domain is located at the C-terminus between residues 325–355 in the human p53 protein (Figure 1). The binding of p53 to specific DNA sequences is regulated by tetramer formation via this tetramerization domain; the C-terminus is also one of the major immunodominant epitopes of human p53 that can be recognized by anti-p53 antibodies in the sera of cancer patients [25,26]. The C-terminal regulatory domain is located at residues 356–393, just following the tetramerization domain (Figure 1). The C-terminal region of p53 is intrinsically disordered and regulated by different post-translational modifications such as phosphorylation, acetylation, methylation, and ubiquitination. The regulatory domain either suppresses or activates p53-dependent transcriptional activities based on the post-transcriptional modifications [27]. MDM2 ubiquitinates p53 at the C-terminal domain, resulting in p53 degradation through the proteasome. Small molecular compound Tenovin is an example of a drug which induces p53 acetylation in this region [28].

The p53 transactivation domains are located near the N-terminus of p53. This domain recognizes and binds to the transcriptional coactivators P300/CBP and components of transcriptional machinery (Figure 1). This N-terminal transactivation domain of p53 binds to MDM2, an E3 ligase [29]. MDM2 ubiquitinates p53 and this results in p53 degradation via the proteasome. MDMX(MDM4) is a homolog of MDM2 but lacks E3-ligase activity; MDMX(MDM4) also binds to p53 TAD and suppresses p53 transcriptional activity. By interacting with MDM2, MDMX(MDM4) stabilizes MDM2 and enhances MDM2 activity [30,31]. Regulation of p53 by MDM2/MDMX(MDM4) keeps basal p53 levels low or undetectable in cells under physiological conditions. The transcriptional domain region of p53 is considered an important domain to target in the development of effective drugs that reactivate p53, which led to the discovery of MDM2 inhibitors such as those in the nutlin family. Most MDM2 inhibitors are not able to significantly inhibit MDMX(MDM4) activity because of the structural differences between MDM2 and MDMX(MDM4) in their p53-binding pockets [31]. Therefore, MDM2 inhibition alone may ineffectively upregulate p53 activity in tumors, especially in the tumors with MDMX(MDM4) overexpression. Dual inhibition of MDM2/MDMX(MDM4) can efficiently release p53 transcriptional activity, which has led to the discoveries of dual inhibitors of MDM2/MDMX(MDM4), such as small-molecule RO-5963 and a stable peptide ALRN-6924 [32,33]. 

Small molecule MDM2/X inhibitors that activate p53 can do so without causing DNA damage, as occurs with cytotoxic DNA damaging chemotherapy or radiation. This domain is modulated by post-translational modifications after DNA damage, such as phosphorylation and acetylation, via cross-talk with many signaling pathways, such as ATM, ATR, and PKI [34]. Phosphorylation of residues clustered in the N-terminal regions regulates the MDM2-p53 interaction as well as p300/CBP. The phosphorylation of residues Ser15 and Ser20 inhibits the p53-MDM2 interaction [35]. There is a second transactivation domain (TAD II) in p53 that is more C-terminal to the N-terminal domain and includes serine 46 and Thr55. The phosphorylation of Serine 46 has been implicated in the activation of apoptosis [36], and the Thr55 phosphorylation inhibits DNA-binding by enhancing competitive interactions between the TAD-II and DNA-binding domain [37]. These studies provide new insight into the design of treatments targeting p53 in combination with other signals to improve the efficacy of cancer therapy.

There is an abundance of p53 isoforms which are truncated either at the N-terminus (such as Δ40-p53, Δ133-p53, and Δ160-p53) or the C-terminus (β-p53 and γ-p53) [38,39]. The p53 isoforms are generated by two p53 transcription promotors (P1 and P2) and alternative splicing of intron 9 or intron 2 in cells [40]. The p53 isoforms lacking the functional domains might contribute to fine-tuning the p53 pathway signaling. For example, Δ40-p53 and Δ133-p53 lack the TAD, and Δ160-p53 lacks the TAD and part of the DBD, but they retain the TD and RD at the C-terminus. Therefore, they can bind to p53 and act through dominant negative effects [41]. The remaining parts of the TAD of the isoforms might regulate different gene expression patterns compared to wild-type full-length p53, therefore possibly displaying GOF [42]. The exact mechanism of action of p53 isoforms in regulating p53 pathway signaling is not completely elucidated. Abnormal expression of p53 isoforms are correlated with aggressive tumor progression [43]. Current strategies targeting p53 are designed mostly based on full-length p53 (wild-type or mutant). It is unclear what is the effect of these approaches on the isoforms. Targeting the isoforms of p53 might be helpful for increasing the efficiency of the therapeutics targeting p53.

### 2.2. Conformational Changes of p53 Mutants Utilized for Drugs That Restore Wild-Type Function 

Most missense mutations in p53 occur in the core domain. The wild-type p53 core domain is unstable with low thermodynamic and kinetic stability, allowing rapid cycling between folded and unfolded states. Mutations in such residues, such as Arg273 and Arg282, destabilize the core DNA-binding domain by enhancing the thermodynamic and kinetic instability of mutant p53. These effects are often exploited to design ligands that selectively bind to the native state of the p53 protein to reverse the thermodynamic and kinetic denaturation consequences of these mutations. 

Mutation at Arg249 of p53 results in impaired DNA-binding due to enhanced flexibility of the L3 loop. Mutation at Y220 causes a cavity formation in the DNA binding core domain that causes loss of p53-DNA binding affinity; these mutations have been exploited in drug development—PhiKan083 (PK083) is a representative example. PK083 is the first small molecule identified to bind to mutant p53 (Y220C), which raises the melting temperature of the mutant and slows down its rate of denaturation [44]. Recently, many Y220C cavity binders have been developed to improve their binding affinities to the Y200C, such as PK9318 [45], MB710, and MB725 [46]. These small molecules increase Tm by 2–4 °C with affinities in the 2–4 μM. The treatment with these Y220C cavity binders restores p53 transcriptional activity and suppresses tumor growth.

Different from other mutations in the core domain, R175H has been described as a zinc-binding mutant. Zinc ions are important in coordinating with C176, H179, C238, and C242 in the core domain to ensure proper folding of the p53 protein [47]. The mutation R175H next to C176 reduces zinc ion binding to the coordinate sites, resulting in mutant p53 protein misfolding, which abolishes DNA binding. Small molecule NSC319726 (ZMC1) plays a role as a zinc metallochaperones in the stabilization of mutant p53-Zn^2+^ interaction, resulting in a wild-type conformational change [48]. 

Mutations at the C-terminus disturb p53 tetramer formation. For example, mutations at Arg337 and Asp352 stabilize the tetramer, and mutation at Arg337 (R337H) disrupts the inter monomer salt bridge, resulting in impaired tetramer formation [49]. However, it is known that not all p53 mutants are equal. The wide variety of mutations in p53 makes it challenging to design drugs that target mutant p53 because different mutations might lead to specific structural changes that may need unique small molecules to fit in.

Promising pharmaceutical strategies using small molecular compounds such as nutlins, ALRN-6924,PRIMA-1, COTI-2, and ZMC1 are being tested as monotherapy or as part of a combination in specific tumor types [50,51,52,53]. 

### 2.3. p53 Transgenic Mouse Models (p53 Knockout, Mutant p53 Knock-Ins, and Inducible Models) Provide a Powerful Tool to Investigate p53 Function 

Cell culture in vitro models mostly used for cancer research provide insights into fundamental biological functions of p53 but do not faithfully reproduce events of tumorigeneses inside living tissues. Mouse p53 is highly homologous to human p53, therefore, p53 transgenic mouse models have been widely applied to explore wild-type and mutant p53 biological functions at physiological conditions and address the mechanism of p53 (wild-type or mutant) action on tumorigenesis and drug resistance in vivo. A broad range of mutant p53 knock-ins or p53 knockout mouse models has been generated. In addition to the constitutive germline-knock-in or knockout of p53, a Cre-LoxP system (Cre acts as a site-specific recombinase to catalyze recombination between specific LoxP sites, and thus depletes genes or gene fragments flanked by LoxP sites) has been applied to conditionally knock in or knock out p53 in mice and to address the role of p53 at different tumor developmental stages [54]. Tissue-specific transcription factor promoters have been exploited for generating tissue-specific expression of Cre, which leads to mutant p53 expression or knockdown of p53 only in the specific tissues [55], therefore, providing an opportunity to investigate the role of mutant p53 or p53 loss in sporadic tumor tissues during tumorigenesis and tumor development. These p53 knock-ins and knockout mouse models mimic initiating events in tumorigenesis and progression of human tumors such as tumor behaviors (tumor growth, metastasis, metabolism, and senescence) and the host immune response in the tumor microenvironment due to p53 loss or p53 mutations.

(1) Studies of p53 using p53-knockout mouse models demonstrate that p53 is a tumor suppressor. The early studies of p53 using p53 germline-knockout mice with different genetic backgrounds, such as 129/sv and C57BL/C, showed that p53−/− mice developed tumors earlier than p53+/− mice. T-cell lymphoma was mostly found in p53−/− mice, while osteosarcomas, soft tissue sarcomas, and lymphoma were the tumors that mostly developed in p53+/− mice [56]. p53 germline-knockout renders mice susceptible to carcinogen-induced tumors [57]. Studies of p53 function using p53 knockout mouse models have demonstrated that p53 loss of function (LOF) increases the incidence of carcinogen-induced tumorigenesis of different tumors [58,59,60,61,62,63]; for example, p53−/− knockout mice are more susceptible than p53+/− or p53+/+ mice to N-methyl-N-nitrosourea stomach carcinogenesis [64]. Conditional knockout of p53 in murine enterocytes is insufficient to initiate intestinal tumorigenesis but markedly enhances carcinogen-induced tumor incidence and leads to invasive cancer and lymph node metastasis [58]. 

It is well known that p53 is a stress sensor and functions as a guardian of the genome by regulating cell cycle arrest and DNA repair or inducing cell death if there is too much unrepaired damage. p53 deficiency leads to genome instability and facilitates the causation of malignant tumorigenesis in response to cellular stresses, such as hyperproliferation, DNA damage, hypoxia, and inflammation signals. Recently, p53-null mouse models showed that the tumor microenvironment could be shaped by p53. p53 loss changes the senescence-associated secretory phenotype (SASP), leading to suppression of the immune response by recruitment of immune inhibitory cells, which promotes tumor growth. For example, pharmacological activation of p53 in vivo unleashed the interferon program, promoted T-cell infiltration, and significantly enhanced the efficacy of checkpoint therapy in a xenograft tumor model [65]. Consistently, p53-null mice showed an accumulation of suppressive regulatory T (Treg) cells in the tumor microenvironment [66]. These fundamental discoveries based on P53-knockout mouse models demonstrate that p53 is a tumor suppressor via different mechanisms of p53 action. Conditional regulation of p53 expression in mouse tissues showed that the restoration of p53 expression suppressed lymphomas and sarcomas growth via cellular apoptosis in different mouse models [67,68]. These studies provide a rationale for the restoration of p53 in cancer therapy, and multiple pharmaceutical approaches to reactivate p53 are under clinical evaluation for targeting therapy and will be discussed in this review.

(2) Mutant p53 knock-in mouse models demonstrate mutant p53 gain-of-function (GOF). p53 missense mutations in the germline were found in patients with Li-Fraumeni syndrome and were associated with cancer development. Patients with Li-Fraumeni syndrome with mutated p53 show tumor onset much earlier than patients with p53 deficiency. p53R248Q/+ patients with Li-Fraumeni syndrome have higher tumor numbers and shorter tumor-free survival by 10.5 years than p53 null/+ patients [69]. This clinical observation was also found in the mice with knock-in of mutant p53R172H (mimic of human p53R175) and mouse p53 R270H (mimic of human mutant p53 R273). These mutant p53 knock-in mice showed greater tumor development and metastases as compared to the p53 null mice in the c57BL/6 and 129/Sv backgrounds, indicating a gain-of-function of mutant p53 [70,71]. p53 mutant knock-in mouse models further support the hypothesis of the unequal functions of p53 mutations. Mutant p53 Knock-in mice harboring the hot spot alleles R248Q and G245S showed their different impacts on tumor development. R248Q/− mice had accelerated onset of all tumor types and shorter survival compared with p53-null mice, but G245S/− mice were similar to null mice in tumor latency and survival, suggesting that mutant p53 GOF alleles are not biologically equal [69]. Conditional knockdown of mutant p53 in cancer cells showed extended mouse survival, and the tumors underwent apoptosis and tumor regression [72]. These studies suggest that tumors are addicted to the sustained high levels of mutant p53. These fundamental discoveries of mutant p53 GOF provide a rationale for depleting mutant p53 in tumors as a strategy for cancer therapy. Targeting mutant p53 degradation by HSP90 inhibitors significantly extended the survival of mutant p53 (R248Q allele2, and R172H allele3) mice as compared to p53−/− littermates [72]. Drug development targeting mutant p53 degradation is a promising strategy for cancer therapy. 

## 3. Strategies for Boosting Wild-Type p53 Activity in Cancer: Gene Therapy, Cytotoxic Chemotherapy, MDM2/MDMX(MDM4) Inhibitors, p53-Binding Compounds, Targeting p53 PTMs 

Chemotherapies used for the treatment of cancer include DNA damaging agents such as doxorubicin, 5-FU, irinotecan, actinomycin D, etoposide, mitomycin D, bleomycin, daunomycin, and cisplatin. These agents induce the DNA damage response and p53, ultimately resulting in apoptosis mediated by p53 target genes. A majority of clinical studies have demonstrated a correlation between adverse clinical outcomes after treatment with chemotherapy in patients with mutant p53-expressing tumors compared to patients with wild-type p53 tumors. Thus, it is thought that wild-type p53 function is at least partly responsible for the clinical efficacy of conventional chemotherapy [73]. Treatments such as conventional chemotherapy and γ-radiation activate p53, which mediates apoptosis through activation of a subset of p53 target genes such as Puma, Noxa, Bax, and death receptor 5 (DR5), among others [74]. Though effective, these conventional chemotherapies induce DNA damage that in some cases leads to secondary malignancies, therefore, novel strategies for specifically targeting p53 are needed [75]. Despite this, there are no FDA-approved therapies that target either wild-type or mutant p53, though some are in various stages of clinical trials [50,76,77]. Advanced strategies, including biotherapeutic and pharmaceutic approaches, have been developed for targeting p53 reactivation for cancer therapy (Figure 2). Biotherapeutic approaches mostly focus on the replacement of p53 with wild-type p53 by gene delivery. Wild-type p53 can be transferred into cancer cells to replace endogenous p53 function using a recombinant virus such as a recombinant adenovirus which fails to replicate efficiently with the E1B-55 Kd protein deletion in cells. Gene therapy based on p53 delivery is under clinical evaluation (Table 1). Pharmaceutical approaches using small molecules for reactivation of wild-type p53 function is a major effort for cancer therapy targeting wild-type p53 (Table 1) [6,30,31].

### 3.1. Small Molecules Can Directly Target Wild-Type p53 Protein 

p53 is a DNA-binding transcription factor that has been characterized as an “undruggable” target, therefore, searching for drugs directly targeting p53 is challenging. Moreover, because the mutations in p53 cause loss of function, directly targeting the protein for functional restoration is challenging. The small molecular compound RITA provides a possibility to directly target p53. RITA was first discovered via cell-based screening in human colon carcinoma lines HCT116 and HCT116 TP53−/− [78]; it directly binds to p53 and interrupts p53 interaction with MDM2, therefore preventing p53 degradation. Further studies showed that RITA treatment leads to a p53-dependent global transcriptional response. RITA suppresses oncogene expression and increases tumor suppressor expression via p53 reactivation, leading to p53-mediated cell death [107]. However, RITA treatment also causes genotoxicity through the integrated stress response (ISR) in wild-type p53 cells as well as accumulated ROS in cancer cells [108,109,110]. ROS-dependent activation of c-Jun N-terminal kinase (JNK) plays a crucial role in RITA treatment-induced p53-mediated inhibition of oncogenes leading to cellular apoptosis [110]. Recent studies show that RITA treatment suppresses tumor growth in p53 defective cancer cells [110,111,112], suggesting non-specificity of RITA in reactivation of p53. 

### 3.2. Small Molecules Can Activate p53 via Targeting MDM2/MDMX(MDM4) 

Inhibition of MDM2 is one of the most promising strategies for targeting wild-type p53 activation. Structure-based screening has led to the identification of a series of small molecules which specifically bind to the p53 protein-binding site in MDM2. Several MDM2 or MDM2/MDMX(MDM4) inhibitors have been developed recently and most of them are under clinical evaluation in different clinical trials (Table 1). Nutlins are a representative example of a class of MDM2 inhibitors. Nutlins are cis-imidazoline analogs and were discovered by biochemical screening methods by the pharmaceutical company Roche in Nutley, New Jersey [113]. Nutlin-3 binds to MDM2 and disassociates MDM2 from p53, thereby rescuing p53 from degradation. In preclinical treatments, the induction of apoptosis by nutlin-3 seems to be specific to wild-type p53. Recently, additional compounds in the nutlin family have been developed which also inhibit MDM2, such as RG7112 [114] and RG7338 [115]. MDM2 inhibitors show specific and potent antitumor effects in various tumors carrying wild-type p53 in preclinical and clinical studies. For example, RG7112 demonstrated clinical activity against relapsed/refractory acute myeloid leukemia (AML) and chronic myelogenous leukemia (CLL). MDM2 inhibition stabilizes wild-type p53 and transcriptionally upregulates gene expression of p53 targets in different tumors such as AML [116,117,118]. In addition to nutlins, AMG232, a selective piperidinone inhibitor of the MDM2-p53 protein–protein interaction, also causes upregulation of p53 signaling and has a potent antitumor effect [119]. Another MDM2 inhibitor DS-3032b (Rain-32/Milademetan) has entered clinical trials for MDM2-amplified cancers, including Merkle cell cancers, among others.

Inhibition of MDM2 and MDMX(MDM4) is considered a potent antitumor approach via activation of p53. MDMX(MDM4) as a paralogue of MDM2 is involved in p53 degradation by augmenting MDM2 activity [120]. Combination treatment with MDM2 and MDMX(MDM4) inhibitors has been investigated since the first MDMX(MDM4) inhibitor SJ172550 was discovered. SJ-172550 binds the p53-binding pocket of MDMX(MDM4) and effectively kills retinoblastoma cells that have high expression of MDMX(MDM4) [121]. Treatment with SJ-172550 combined with MDM2 inhibitors additively suppresses tumor growth [121]. Similar results were also observed after combination treatment with nutlins and XI-011 (NSC146109), an MDMX(MDM4) inhibitor [88,104], suggesting that dual inhibition of MDM2/MDMX(MDM4) causes a more potent antitumor effect and activation of p53 compared to treatment with either alone. Compounds that dually inhibit MDM2 and MDMX(MDM4) have been developed, such as ALRN-6924 [33]. ALRN-6924 is a cell-penetrating stapled α-helical peptide that robustly activates p53-dependent transcription by preventing its interaction with MDM2/MDMX(MDM4) and induces cell cycle arrest and apoptosis in TP53 wild-type tumors [33]. Clinical evaluations at phase I show that ALRN-6924 is well tolerated and demonstrates anti-tumor activity [122].

### 3.3. Small Molecules Can Induce p53 Transcription via Post-Translational Modifications 

Post-translational modification is one of the key steps for p53 activation as a transcription factor. The small molecule tenovins was identified as a p53 activator via acetylation of p53. Tenovins blocks deacetylation of SirT1 substrates, including p53, and enhances p53-dependent transcription in cells through SirT1 inhibition [28]. Phosphorylation of p53 at different sites in the C-terminus affects p53 functions. The small molecule curaxin phosphorylates p53 at Ser392 through casein kinase 2 [123]. TopIn treatment results in phosphorylation of p53 at Ser15 and stabilizes the p53 protein by dissociation of MDM2 and p53 [124]. These studies provide a rationale for therapeutic targeting of post-translational modification of p53 to activate its transcriptional function. It is recognized that there is potential for non-specific effects of these small molecular weight compounds since these kinases are not p53-specific. Even so, upregulation of p53 via post-translational modifications can be exploited in combination with other therapies which target other parts of the p53 regulatory network to increase the efficacy of p53-targeting therapy. 

## 4. p53-Dependent and p53-Independent Strategies for Targeting p53 Pathway Restoration in p53-Mutated Cancers

The tumor-specific mutant p53 is an attractive target candidate for cancer therapy development. Small molecules have been used to restore p53 function and inhibit mutant p53 GOF (Figure 2). These pharmacological approaches have been investigated in different tumors carrying mutant p53. In addition, biotherapeutics appear to be potentially powerful approaches for targeting mutant p53 in cancer cells. CRISPR is a representative example of genome modification of mutant p53. Targeting mutant p53 genomic modifications by CRISPR is at its beginning and may soon be able to clinically target mutant p53 for cancer therapy. In this review, however, we focus on pharmacological approaches for targeting mutant p53 in cancer therapy.

### 4.1. Restoration of Wild-Type Function in Tumors Expressing Mutant p53

Restoration of wild-type function in mutant p53-expressing tumors is a promising strategy for cancer therapy; however, restoration of wild-type function of p53 is challenging because of the high variation of p53 mutations in cancer. PRIMA-1 (APR-246) is a representative example of a p53-restoring compound. PRIMA-1 (APR-246) is converted to the reactive electrophile methylene quinuclidinone (MQ), which covalently binds to Cys124 and 277 at the core domain of mutant p53. The binding of MQ causes refolding of mutant p53 and restores wild-type functions [86,125,126]. Similar thiol reactivity of mutant p53 was also found with other mutant p53-reactivating compounds, such as MIRA-1, CP-31398, STIMA-1, and 3-benzoylacrylic acid (Table 1). They bind to mutant p53 by Michael addition. Restoration of wild-type p53 by p53-reactivating small molecular weight compounds such as PRIMA-1 (ARP-246) induces p53 target gene expression (such as p21, Puma, and Noxa), triggers cell death, and suppresses tumor growth in vivo. APR-246 is the first p53-restoring compound under clinical evaluation either as monotherapy or in combination treatment. APR-246 in combination with azacytidine is under clinical evaluation at phase I and III in acute myeloid leukemia. 

Phikan083 and SCH529074 are two small molecules identified via structure-based assays. Both compounds restore wild-type p53 function to mutant p53, but the mechanism is different. Phikan083 is a carbazole derivative identified based on a docking-based screen for a molecule that fits into this cavity. Phikan083 binds to the cavity that results from the Y220C mutation of p53 and restores wild-type function [44]. By contrast to Phikano83, SCH529074 acts as a chaperone and binds specifically to the p53 DNA-binding domain, restoring DNA-binding activity to mutant p53 [127]. A single amino acid change (N268R) abolishes the binding of SCH529074 [127]. Alkylating drugs 3-benzoylacrylic acid and its fluorinated derivative (E)-4-(4-fluorophenyl)-4-oxobut-2-enoic acid were identified to covalently bind to the p53 binding domain containing the mutations Y220C, R175H, G245S, R249S, or R282 by increasing the melting temperature of the core domain [90,92]. 

NSC319726 (ZMC1) is a small molecule zinc metallochaperone that specifically refolds mutant p53 (R175) to the wild-type conformation [48]. ZMC1 facilitates zinc ion binding to mutant p53 and leads to wild-type-like conformation changes, restoring wild-type-like functions. ZMC1 upregulates p53 target expression and induces cellular apoptosis in cancer cells harboring the R175 mutation. In addition, a novel thiosemicarbazone derivative COTI-2 was found to bind to different p53 mutants and wild-type p53. COTI-2 induces the conformational change of mutant p53 (including R245 and R175) to wild-type, therefore, restoring p53 transcriptional activities [14,93]. COTI-2 might likely work as a thiosemicarbazone metal ion chelator to induce a conformational change of mutant type p53 with the wild-type. However, unlike ZMC1, COTI-2 showed no effects on intracellular zinc levels in HNSCC-mutant TP53 cells, suggesting that COTI-2 is not a traditional zinc metallochaperone [14]. Current studies showed a potential antitumor efficacy of COTI-2 in different tumor types through both p53-dependent and p53-independent mechanisms [14,128]. COTI-2 treatment affects other targets that are independent of p53 activates tumor suppressor AMPK, and inactivates the oncogene mTOR in HNSCC [14]. 

### 4.2. Therapeutic Induction of Mutant p53 Degradation

Mutant p53 is stabilized in most cancer cells. Cancer cells are addicted to mutant p53 for their survival and proliferation. Preclinical studies showed that knockdown of mutant p53 extended mouse survival and repressed tumor growth [72], suggesting that depletion of mutant p53 is a promising therapeutic approach. However, genomic depletion of mutant p53 is not yet available for cancer therapy in the clinic. Most studies are focused on pharmacological approaches to induce mutant p53 degradation via the proteasome or lysosome. 

One example of a group of compounds that exploit the proteasome to induce mutant p53 degradation are HSP90 inhibitors, such as geldanamycin [129] and 17AAG [130]. HSP90 inhibits MDM2, therefore, protecting mutant p53 from MDM2-mediated ubiquitination [131]. Inhibition of HSP90 activity releases MDM2 from HSP90 and allows MDM2 to interact with mutant p53. HDAC inhibitor, SAHA, inhibits HSP90 activity, thus resulting in mutant p53 degradation in cancer [132]. These studies suggest that MDM2 is a key factor in mutant p53 destabilization. The role of MDM2 in mutant p53 degradation was exploited by NSC59984 to induce mutant p53 degradation in human cancer cells. NSC59984 is a small molecule that induces mutant p53 degradation via activation of MDM2 [21]. In addition to MDM2-mediated mutant p53 degradation via the proteasome, HSP40-CHIP appears to be a new pathway involved in mutant p53 degradation. Small molecules statins (cholesterol-lowering drugs) were found to induce mutant p53 (R175) degradation via HSP40-CHIP [101]. These studies provide a potential strategy to deplete mutant p53 through a mevalonate pathway–HSP40 (DNAJA1) axis.

Spaurtin is a representative example for exploiting the lysosome to induce mutant p53 degradation through the chaperone-mediated autophagy (CMA) pathway [102]. These small molecules induce mutant p53 degradation via interruption of p53 stabilization. It is unclear whether they directly bind to mutant p53. The identification of CMA as a new degradative mechanism for mutant p53 provides the possibility of activating CMA as a new treatment for cancers with mutant TP53. 

Recently, proteolysis-targeting chimeras (PROTACS) have been developed to modulate protein degradation. PROTACs are heterobifunctional molecules consisting of one ligand for binding to a protein of interest and another to an E3 ubiquitin (E3) ligase, connected via a linker [133]. This promising technique will provide a new approach for drug discovery by targeting mutant p53 degradation. 

### 4.3. Activation of p73 to Bypassing Mutant p53 in Restoration of Wild-Type p53 Function 

Mutant p53 forms an inhibitory complex with p73 to abolish the transactivation and proapoptotic activity of p73. p73 is a member of the p53 family and has a similar structure and function to p53. Active p73 can induce cellular apoptosis via the regulation of p53-targets. Our lab conducted an early HTS and found that a significant proportion of the selected candidates relied on p73 for p53-reporter activation in p53 mutant cancer cells, providing a rationale for activating p73, thereby bypassing mutant p53 loss-of-function [20]. 

The interruption of mutant p53 binding to p73 results in the release of p73 from the inhibitory complex. Small molecular compound RETRA is an example of a compound that interrupts mutant p53 GOF. RETRA binds to mutant p53, blocks interaction between mutant p53 and p73, and results in p73 activation [103]. Similar results were also found in treatment with prodigiosin. Prodigiosin promotes p53 pathway activity via releasing p73 from the inhibitory complex with mutant p53 in p53 mutant-cancer cells [22]. RETRA and prodigiosin produce tumor-suppressor effects through p73-like reactivation of p53. 

NSC59984 also can activate the p73 pathway by degradation of mutant p53, which is induced via a ROS–ERK2–MDM2 axis [21,134]. Different from NSC59984, statins were not found to upregulate p73 signaling, though they can induce mutant p53 degradation [101]. These studies suggest that the release of p73 from the mutant p53 inhibitory complex is one of the steps for p73 activation. p73 activation may need further regulation through multiple signaling pathways. Targeting p73 activation in combination with depletion of mutant p53 GOF or small molecular weight compounds with the dual effect could be a promising approach for a potent antitumor effect. 

### 4.4. Restoration of p53 Pathway Signaling Independent of p53 through Non-Canonical Regulatory Pathways

Global gene expression panels show multiple genes transcriptionally regulated by p53 in a canonical way [135,136,137,138,139]. In addition, p53 pathway signaling was found to partially overlap with the other signaling pathways. It is known that the canonical p53 targets can be regulated via other factors via a p53 independent manner [140,141,142,143]. For example, p21 is a p53 transcriptional target and transcriptionally upregulated via p53 binding to its promoter. In addition, p21 also is regulated via other factors either at the transcriptional level or post-translational modifications independent of p53 [144]. Non-canonical pathways also regulate p53 pathway signaling in addition to the canonical p53 regulation. The proof concept of non-canonical regulation of p53 pathway signaling was raised in our recent investigations of p53 pathway restoration. The transcriptomic and proteomic analysis showed that the compounds PG3-Oc, CB002, and their analogues increased ATF3/4 transcriptomic and proteomic targets which overlapped with canonical p53 pathway signaling [145,146]. The integrated stress response ATF3 and ATF4 transcription factors play a role in the stimulation of pro-apoptotic p53 targets by the p53 pathway restoring compounds in tumor cells expressing mutant p53. We found that PG3-Oc partially upregulates the p53-transcriptome (13.7% of public p53 target-gene dataset; 15.2% of the in-house dataset) and the p53-proteome (18%, HT29; 16%, HCT116-p53−/−). ATF4 is a key regulator of PG3-Oc-induced p53 pathway restoration. ATF4 shares a subset of p53 target genes involved in cell cycle arrest and apoptosis, such as PUMA, DR5, p21, Noxa, and NAG-1 in cells in response to PG3-oc [145]. We detected 197 genes in the Fischer p53 dataset (343 genes) and out of those, 102 were differentially expressed by a CB002 analogue in p53 deficient cancer cells, indicating that nearly 50% of the established p53 target genes were altered [146]. Our investigation suggests that ATF4 is a non-canonical factor in restoring p53 pathway signaling independent of p53. ATF3/4 was found to induce NOXA expression by binding to NOXA promoter in head and neck squamous cell carcinoma (HNSCC) in response to cisplatin treatment [141]. These results suggest that p53 pathway signaling can be regulated via non-canonical regulatory pathways (Figure 3). These non-canonical regulation pathways compensate for p53 loss and play an important role in restoring p53 pathway signaling in tumor cells that express mutant p53. The p53 canonical targets Noxa and Puma are regulated by ATF3/4 via non-canonical regulatory pathways and are required to induce cell death by CB002 and PG3-Oc, respectively [145,146]. These results provide a therapeutic strategy for modulating components of non-canonical regulatory pathways to achieve partial restoration of the global p53 transcriptome and proteome that include critical p53-effectors of cell cycle arrest, apoptosis, and metastasis in p53-deficient cancer cells.

## 5. Targeting p53 Function in Immunotherapy: Bispecific Antibodies, Gene Therapy, Small Molecule Combinations 

Immunotherapy is a revolutionary development for cancer therapy [147]. Mutant p53 and wild-type p53 both relate to the immune response. p53 function related to the immune response provides a possibility for combination treatments with immune checkpoint blockade (ICB) in different tumors. With high protein levels and tumor specificity, mutant p53 becomes one of most relevant tumor antigens to target with tumor vaccines and immunotherapy. 

### 5.1. Activation of Wild-Type p53 for Immunotherapy Using Immune Checkpoint Inhibitors

Activation of p53 promotes T-cell infiltration and sensitizes tumor cells to checkpoint therapy. There are some successful examples of activation of p53 in combination with immune checkpoint blockade (ICB) in different tumors. rADp53 in situ gene therapy in combination with pembrolizumab has shown a potent antitumor effect and prolonged survival in different tumor models [148]. The P53MVA vaccine activates CD8+ cells [149] and the P53MVA vaccine in combination with pembrolizumab has appeared feasible, safe, and may offer clinical benefit in patients [150]. Similar to the overexpression of wild-type p53 such as rADp53 and P53MVA, MDM2/MDMX(MDM4) dual inhibitor ALRN-6924 shows that activation of p53 boosts the immune response against tumors by activating interferons in mouse models and human patients [151]. p53-restoring compound PRIMA-1 (APR-246) in combination with pembrolizumab is under clinical evaluation in phase I/II. 

### 5.2. Targeting Mutant p53 in Immunotherapy

CD8+ and CD4+ T-lymphocytes recognize short peptides (epitopes as tumor antigens) derived from proteins presented on the cell surface in association with class I or II human leukocyte antigen (HLA) molecules. High expression of mutant p53 provides a possibility to generate vaccines to prevent tumor growth. However, mutant p53 is localized inside of cells, not on the cell surface. 

Recent studies detected mutant p53 peptides presented on the cell surface with HLA complex. This mutant p53 peptide-HLA complex on the cancer cell surface provided a possibility for designing a new approach for immunotherapy by enhancing T-cell recognition of cancer cells based on these neoantigens. For this purpose, Vogelstein and his colleagues developed a new bispecific antibody H2-scDb, with one arm binding to a T-cell receptor and the other recognizing the mutant p53 (R175) peptide-HLA complex on the cancer cell surface [104]. This bispecific antibody function acts as a bridge between cancer cells and T-cells and stimulates T-cells to kill tumor cells. Treatment with this bispecific antibody specifically targets mutant p53-expressing tumor cells and significantly suppresses tumor growth. The bispecific antibody generated with different p53 mutations will bring T cell-based immunotherapy one step forward toward the development of precision cancer medicine.

## 6. Combination Therapies Targeting p53 Cell Cycle Checkpoints, MDM2-p53 Inhibition, or Mutant p53

Multiple oncogenic and signaling pathway abnormalities in individual cancers promote cancer cell resistance to conventional chemotherapy. Combinatorial treatment with two different drugs with individual targets can enhance antitumor efficacy compared to a monotherapy approach. This approach is widely used in cancer therapy in the clinic [152]. Clinical studies show the effect of AdP53 gene therapy combined with chemotherapy or radiotherapy. These studies provide a rationale for targeting p53 reactivation in combination treatments. p53 activation in combination treatment improves antitumor efficacy and minimizes side effects and newly developed resistance in tumors at different progressive stages.

### 6.1. Targeting p53 in Combination with Cell Stress Signaling 

ROS stresses are common events in cancer cells. Administration of p53-restoring compounds often results in cell stress and increases cellular ROS levels. Cellular stress has been exploited in combination treatment with p53-targeting compounds. PRIMA-1 (APR-246) is one example of a p53-reactivating compound that induces cellular stress that can be exploited with combination treatments. PRIMA-1 (APR-246) was found to increase ROS by depleting GSH. SCL7A11 is a component of the xC(-) system in the regulation of GSH and is considered a major biomarker for APR-246 treatment in cancer [153,154]. PRIMA combined with an xC(-) system inhibitor, which increases ROS, promotes the antitumor effect of the combination treatment [155]. 

Signaling pathways related to cellular stresses can be targeted in combination with p53-reactivating compounds. JNK is a signaling pathway related to the stress response and has been found to be activated in the p53 pathway targeting treatments; the p53 restoring compound PRIMA was found to induce p53-dependent apoptosis through this pathway in colon cancer [156]. Treatment with the p53-reactivating compound RITA demonstrated a significant number of differentially expressed genes associated with the stress response, including the JNK signaling pathway. Combined treatment with RITA and a JNK activator synergizes in cytotoxic responses in MM cell lines and patient samples [157]. The mechanism by which small molecules induce p53-mediated apoptosis through the JNK signaling pathway provides a rationale for the combination of p53 activating drugs with JNK activators in the treatment of some types of cancer. These studies propose a new approach for targeting mutant p53 in combination with cellular stress signaling.

### 6.2. Targeting p53 in Combination with Inhibition of Raf/MEK/ERK Pathways 

Abnormalities in p53 and the Raf/MEK/ERK axis are among the most common during tumorigenesis and tumor development in various cancers. Targeting p53 in combination with inhibition of the Raf/MEK/ERK pathway is effective in certain drug-resistant cancer cells. The simultaneous blockade of MEK and MDM2 signaling by nutlin-3a triggered synergistic, pro-apoptotic responses in AML cell lines [158]. Similar results were also observed with combination treatment of MDM2 antagonist (RG7388) and MEK inhibitors in dedifferentiated liposarcoma [159]. In melanoma cells, treatment with nutlins combined with inhibitors of ERK2 was found to synergistically induce cell apoptosis. Nutlin-3 combined with vemurafenib (inhibitor of mutant Raf) synergistically induced apoptosis and suppressed melanoma cell viability in vitro and tumor growth in vivo, suggesting that reactivation of p53 by nutlins overcomes vemurafenib resistance in melanoma [160]. Sorafenib, a Raf inhibitor, along with nutlin-3, synergistically induced apoptosis in RCC [161]. 

These results strongly indicate the therapeutic potential of combined MEK blockade and MDM2 inhibition to activate p53 in cancer cells. A p53-based combinatorial approach can reduce doses and side effects of agents in combinations and efficiently suppresses tumor growth.

### 6.3. Targeting p53 (Wild-Type and Mutant) in Combination with Conventional Chemotherapy

Clinical studies demonstrate improved outcomes after treatment with conventional chemotherapy in patients with tumors harboring wild-type p53 compared to mutant p53, suggesting a crucial role of wild-type p53 in the cellular response to chemotherapy. This idea is supported by the observation that lymphoid cells from p53 knockout mice experience complete chemoresistance [162]. Furthermore, others have observed that wild-type p53 is necessary for cytotoxicity of conventional chemotherapy in vitro [163,164,165]. These findings suggest that overexpression or reactivation of wild-type p53 may synergize with chemotherapy treatment to induce apoptosis in cancer cells, and this has indeed been observed in vitro [166,167]. 

p53-targeting small molecules have been investigated for an antitumor effect not only as a monotreatment, but also in combination with chemotherapy. Conventional chemotherapeutic agents 5-FU, doxorubicin, etoposide, and cisplatin are often used in combination treatment with p53-targeting agents. p53-based chemotherapy significantly reduces the genotoxic burden and induces high anti-tumor efficacy.

Nutlins have been widely used in treating cancer cells with wild-type p53. The effectiveness of nutlin-3 in combination with chemotherapy has been observed in preclinical studies. Nutlins and their analogs (such as MI-43, MI-63, and MI-319) synergize with genotoxic drugs such as 5-FU, doxorubicin, etoposide, and cisplatin in multiple tumors carrying wild-type p53 such as hematological malignancies, lymphoma, neuroblastoma, hepatocellular carcinoma, lung cancer, and pancreatic cancer. Nutlins in combination with chemotherapy show minimal toxic effects in normal cells. Dual inhibition of MDM2/MDMX(MDM4) by ALRN-6924 enhances antitumor efficacy of chemotherapy in TP53 wild-type hormone receptor-positive breast cancer models [51]. RITA, in combination with doxorubicin, enhanced the sensitivity of NALM-6 cells to doxorubicin and promoted doxorubicin-induced apoptosis. Small molecules targeting p53 activation decrease the necessary dose of conventional chemotherapy to reduce genotoxicity but enhance or magnify the effects of the chemotherapies. 

Targeting mutant p53 in combination with conventional chemotherapy is another approach for increasing antitumor efficacy. Mutant P53 is a contributing factor for drug resistance. p53 mutations have been reported to correlate with resistance to platinum chemotherapy in different types of cancer cells. PRIMA-1 (APR-246) is one of the p53-restoring compounds that is administered in combination treatment with conventional chemotherapy. Combination treatment with PRIMA-1 (APR-246) overcomes cancer cell resistance to chemotherapy. PRIMA-1 was found to synergistically induce cell death in combination with multiple chemo-drugs such as in combination with adriamycin in NSCLC cell lines carrying different p53 mutations [168] or in combination with doxorubicin or cisplatin in thyroid cancer cells [169]. APR-246 has been reported to overcome chemoresistance to cisplatin and doxorubicin in p53-mutant ovarian cancer cells [170]. Combination treatment based on targeting the mutant P53 pathway prevents the development of drug resistance in cancer cells. 

The synergistic effect of p53 overexpression or reactivation and chemotherapy provides a promising new direction for the development of combination therapies for patients with tumors containing mutant p53.

### 6.4. Combination Targeting p53 Cell Cycle Checkpoints in p53 Mutated Cancer Cells 

p53 is involved in the G1-S cell cycle checkpoint via regulation of p21 and arrests cells at the G1 phase to allow enough time for DNA repair or otherwise induces cell death in response to DNA damage. Mutant p53-expressing cancer cells have defective G1 cell cycle checkpoints, and this causes such cancer cells to be more reliant on alternative mechanisms to maintain cell survival. The ataxia telangiectasia and Rad3-related kinase (ATR)- checkpoint kinase 1 (Chk1) axis and WEE1 are compensatory pathways independent of p53 involved in intra-S and G2/M checkpoints in response to DNA damage [171]. These cellular vulnerabilities enforced by mutant p53 can be exploited by further targeting of the checkpoints in response to DNA damage in p53-mutated cancer cells. Accumulating studies show that inhibition of ATR/Chk1/WEE1 sensitizes p53-deficient cancer cells to DNA-damaging agents. 

Chk1 has a critical role in the DNA damage response (DDR) and cell cycle checkpoints at the S phase and at the G2/M transition [172]. Studies based on in vivo and in vitro experiments show that Chk1 inhibitors (such as UCN-01, AZD7762) sensitize p53-deficient cancer cells to DNA-damaging agents (such as irinotecan and cisplatin) in different tumor types, including TNBC, head and neck cancer, and pancreatic cancer [173,174,175]. The inhibition of Chk1 potentiates the cytotoxicity of DNA damaging agents in p53 deficient cells. 

WEE1 is a crucial regulator of the G2/M checkpoint of the cell cycle independent of p53 [176]. Abrogation of the G2/M checkpoint by WEE1 inhibition can sensitize p53-deficient cells to DNA-damaging agents. This is supported by results from experiments with WEE inhibitors in combination with chemotherapy and radiation therapy in p53 mutant cancer cells. MK1775 (Adavosertib, AZD1775) is a small molecular weight WEE1 kinase inhibitor. Blockade of WEE1 by the inhibitor MK1775 synergistically induces cell death in p53 mutant colonic cancer when combined with irinotecan [177]. MK-1775 also synergizes with gemcitabine to suppress tumor growth selectively in p53-deficient pancreatic cancer and ovarian cancer [178,179]. MK1775 prevents WEE1-mediated phosphorylation CDK1. 

WEE1 regulates CDK1 activity at the G2/M checkpoint and prevents entry into mitosis. Inhibition of WEE1 causes mitotic entry without completion of DNA repair and replication upon DNA damage, resulting in mitotic catastrophe and apoptosis. The combination results in significant DNA damage due to failure of DNA repair in the cells, which lose both the p53-checkpoint at G1 and the WEE1-checkpoint at G2, leading to synthetic lethality. Clinical studies at phase II show a promising antitumor efficiency of the combination of MK1775 with chemotherapy such as carboplatin and paclitaxel in p53-mutated ovarian cancer [180,181].

Many ATR, Chk1, and WEE1 inhibitors have been developed and evaluated in different tumors in clinical trials [182,183]. Mutant p53 can be a synthetic lethality factor in cancer cells to increase antitumor efficacy of ATR/Chk1/WEE1 inhibitors targeting checkpoints in combination when combined with chemotherapy for cancer therapy. 

## 7. Summary and Prospects

The tumor-specific p53 status and the important role of p53 in the regulation of cell function indicate p53 as a promising target for cancer therapy. Although there are no FDA-approved drugs targeting p53 (wild-type or mutant), advanced strategies based on p53 structure and function have continued to be developed for targeting p53 in cancer therapy (Figure 2 and Figure 3), and some of them are under clinical evaluation (Table 1). Among the strategies, MDM2 is considered one of the key factors in drug targeting wild-type p53 or mutant p53. Interruption of p53 binding to MDM2 is one of the promising strategies to reactivate p53. This has led to the identification of MDM2 inhibitors. By contrast, inducing MDM2-mediated ubiquitination of mutant p53 is another promising strategy to induce mutant p53 degradation, and this has pointed to HSP90 inhibitors, HDAC inhibitors, and NSC59984 that activate MDM2 and destabilize mutant p53. Though clinical trials have demonstrated that MDM2 inhibitors have anti-tumor activity and acceptable safety profiles, some limitations exist, including the development of resistance and dose-limiting toxicity [184,185]. Common resistance mechanisms include the emergence of p53 mutations in tumors that were previously wild-type for p53, alteration of various oncogenic pathways, and MDMX(MDM4) overexpression [186]. Sequestration of wild-type p53 by MDMX(MDM4) after MDM2 inhibition has sparked interest in dual MDM2/MDMX(MDM4) inhibition [187]. Several clinical trials are investigating dual MDM2/MDMX(MDM4) inhibition, but none have been approved yet (Table 1). Further refinement of desirable effects of these small molecules will facilitate the translation of these promising compounds into clinical practice. Non-specific and off-target effects are a common weakness of small molecules when they are applied in cancer therapy. To improve the efficacy and reduce the off-target effects, targeting p53 in combination treatment has been widely investigated in preclinical and clinical research as a promising strategy for enhancing antitumor efficacy and reducing toxicity. Searching potential biomarkers will be essential for targeting p53 in combination therapy. Next genomic sequencing and proteomic assays provide global gene and gene expression patterns of tumors derived from patients. This information may be useful for rationally designing therapeutic approaches by targeting p53 and/or in combination with other altered signal pathways as additional targets in cancers. The patient-derived tumor organoids or patient-derived xenografts mouse models may become invaluable systems to further identify therapeutic approaches and contribute to personalized cancer medicine.

Besides the pharmacological approaches currently developed, biotherapeutic approaches targeting p53 (such as gene therapy and genomic editing) appear as promising strategies in cancer therapy. p53 based gene therapy has been considered a promising approach for cancer therapy as a monotherapy or in combination with chemo- and radiotherapy in a range of tumor types in clinical trials in China [83,188]. Selective delivery of wild-type p53 can increase the specificity of p53 gene therapy. Recently, cationic liposomes coated with anti-transferrin receptor single-chain Ab fragment (scL) has been used to help deliver wild-type p53 (scL-53). This scL53 nanocomplex can be specifically delivered to tumors based on the tumor transferrin receptor that is not expressed on normal cells [189]. The scL serves as an example and provides a new approach for p53-based gene therapy with high specificity and low toxicity. Genetic editing of mutant p53 appears to be an attractive strategy for p53-targeting therapy. The CRISPR/Cas9 system has been used to repair the TP53 414delC mutation to the wild-type TP53 genotype and inhibit the cell proliferation of PC-3 cells [190]. The CRISPR/Cas9 system could become a powerful technique that will bring genomic editing of p53 closer to cancer therapy [191,192].

## Figures and Tables

**Figure 1 biomolecules-12-00548-f001:**
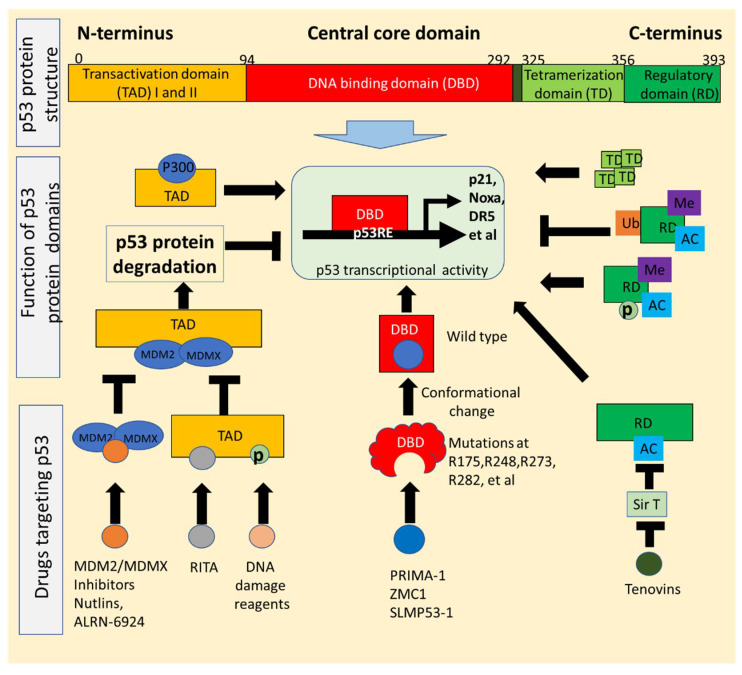
p53 functional domains are utilized for drug development. Wild-type p53 protein contains several functional domains located at the N- and C terminal regions and the central core domain. The transactivation domain (TAD) is recognized and bound by transcriptional coactivators or MDM2/MDMX(MDM4). MDM2/MDMX(MDM4) binds to the TAD and sequentially ubiquitinates the C terminal region, resulting in p53 degradation. Blockade of p53- MDM2/MDMX(MDM4) interaction results in upregulation of p53 transcriptional activity through (1) post-translational modification (PTM) of the TAD such as phosphorylation at Ser15 and Ser20 in response to DNA damages, (2) small molecule occupies MDM2-binding sites at the TAD, such as RITA, or (3) inhibition of MDM2/MDMX(MDM4) activity with MDM2 inhibitors or MDM2/MDMX(MDM4) dual inhibitors. The DNA binding domain (DBD) binds to the consensus DNA sequences and regulates specific gene expression. Small molecules convert the mutated DBD to the wild-type, therefore restoring wild-type p53 transcriptional activity. The tetramerization domain (TD) is critical for p53 tetramer, which stabilizes p53DBD binding to the consensus DNA sequence. The C-terminal regulatory domain is involved in regulating p53 transcriptional activity by either suppressing or enhancing the DBD binding to the consensus DNA based on specific PTMs. Targeting specific post-translational modifications of the RD was exploited to identify tenovin-1 as a p53 activator via p53 acetylation at lysine 382. AC, acetylation; Me, methylation; P, phosphorylation; Ub, ubiquitination.

**Figure 2 biomolecules-12-00548-f002:**
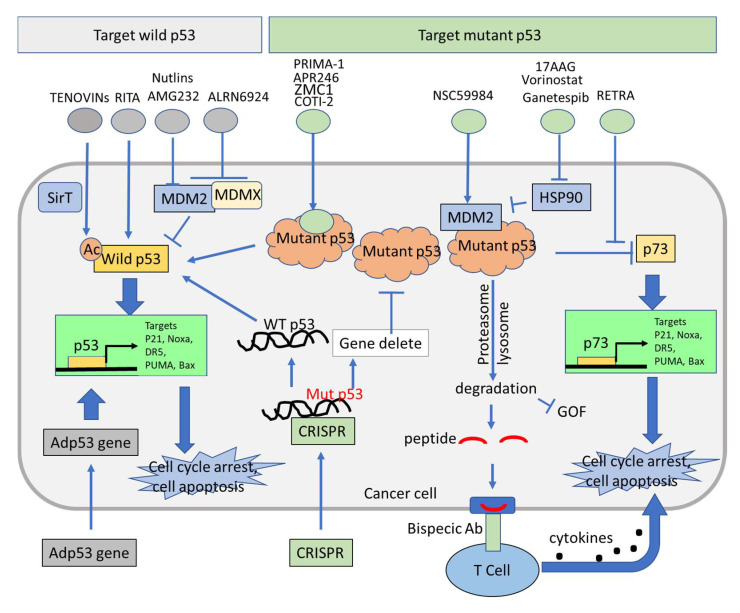
Strategies for targeting mutant p53 and wild-type p53 in cancer cells. Pharmacological approaches for targeting wild-type and mutant p53 in cancer cells are focused on small molecules (upper panel). Small molecules targeting wild-type p53 activation via binding to p53 (such as RITA), inhibition of MDM2/X (such as an MDM2 inhibitor nutlin-3 and the dual inhibitor ALRN6924), post-translational modifications (such as tenovin). Small molecules target mutant p53 via restoration of p53 function (such as PRIMA-1), degradation of mutant p53 via activation of MDM2 (such as 17AAG and NSC59984) or interruption of mutant p53-p73 interaction (such as RETRA). Activation of p73 upregulates p53 target gene expression and induces cell death. Biotherapeutic approaches are based on gene transfection and genomic modifications (bottom panel). p53 is transfected into cancer cells with an adenovirus to replace mutant p53, and upregulates p53 signaling (such as rADp53). Genomic editing is used to restore wild-type p53 or delete mutant p53 in cancer cells by genome editing approaches (such as CRISPR). A bispecific antibody with mutant p53-specific peptide and ALH ligands promotes T cells to recognize and kill p53-mutant tumor cells in cancer immunotherapy.

**Figure 3 biomolecules-12-00548-f003:**
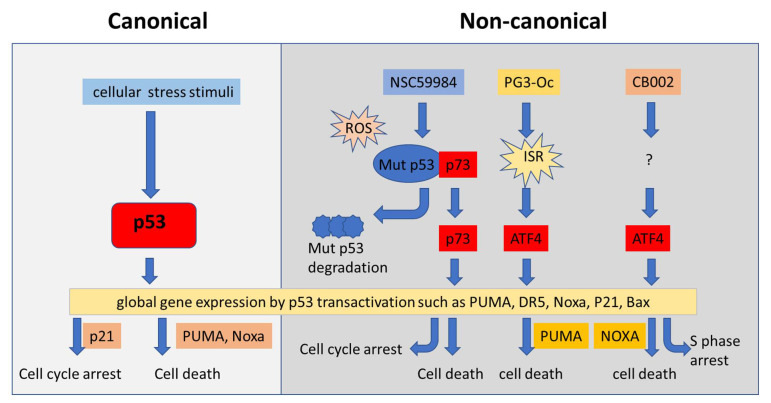
Restoration of p53 pathway signaling in canonical and non-canonical regulatory pathways in cancer cells. Cellular stresses activate p53 pathway signaling through p53 transactivation (canonical pathway, left panel). The p53 pathway signaling is regulated via p73, a p53 family member (such as by NSC59984), or via ATF4 (induction by small molecules such as PG3-Oc and CB002) in mutant p53-expressing cancer cells (non-canonical regulatory pathways, right panel). The gene expression via ATF4 or p73 partially overlaps with the canonical p53 targets, and some of the overlapping canonical p53 pathway signaling target genes (such as Noxa, PUMA, and p21) are crucial for the compounds to induce cell death and cell cycle arrest under different cellular conditions.

**Table 1 biomolecules-12-00548-t001:** Overview of mutant-p53 targeting and wild-type p53 activating agents.

Compound/Peptide/Antibody			Chemical Name and/or Class	Target/Mechanism	Clinical Development	References
Targeting wild p53 activation	p53 activator	RITA		Binds to p53 and prevents WT p53 degradation by blocking interaction with MDM2	Experimental and/or preclinical	[78]
	MDM2 inhbitors	Nutlin-3a	Cis-imidazoline	Blocks the interactive binding sites of p53 and MDM2, dramatically increasing the half-life of p53 and activating p53-mediated transcription.	The listed inhibitors, except nutlin-3a, have undergone or are currently undergoing clinical trials	[9,79]
		RG7112	Cis-imidazoline			
		RG7388	Cis-imidazoline			
		RG7775	Pegylated prodrug idasanutlin			
		MI-77301	Spirooxindole			
		AMG232	Piperidinone			
		SAR405838	Piperidinone			
		MK-8242	2(1H)-Pyrimidinone			
		CGM097	Dihydroisoquinolinone			
		DS-3032b	Unknown			
		HDM201	Imidazopyrrolidinone			
	MDM2/MDMX(MDM4) dual inhibitors	ALRN-6924	Stapled peptide	Blocks the interactive binding sites of p53 and MDM2/MDMX(MDM4), dramatically increasing the half-life of p53 and activating p53-mediated transcription.	currently undergoing clinical trials	
		RO-5963 RO-2443	indolyl hydantoin	Binds to MDMX(MDM4)/MDM2 and blocks p53-MDM2/MDMX interaction		[32]
	MDM2 degradators	PROTAC 8, A1874	IMiD-based MDM2	Targeted degradation of MDM2 using proteolysis targeting chimeras (PROTACs)	Experimental and/or preclinical	[9,80]
	Gene therapy- based on oncolytic Viruses	ONYX-015	Recombinant adenovirus with wild-type p53 (Ad-p53)	A mutant adenovirus with a deleted E1B-55Kd gene commonly fails to replicate efficiently in cells with a wild-type p53 but replicates in many (but not all) cells with a mutant p53 gene.	In clinical trials	[9,81,82]
		Gendicine (Ad-53)	Recombinant adenovirus engineered to express wildtype-p53 (rAd-p53)	Gene replacement (gene therapy)	Approved in 2003 by the China Food and Drug Administration (CFDA) to treat head and neck cancer	[9,83,84]
Targeting mutant p53	Restoration of wild-type function to mutant p53	CP-31398	Styrylquinazoline	Cysteine-binding compounds, Michael acceptor binding to mutant p53	Experimental and/or preclinical	[85]
		PRIMA-1	Quinuclidinone	Cysteine-binding compound is converted to MQ, which binds mutant p53 by Michael addition	Experimental and/or preclinical	[86]
		APR-246	Quinuclidinone	Cysteine-binding compound is converted to MQ, which binds mutant p53 by Michael addition	Phase Ib/II for ovarian cancer, MDS, and oesophageal cancer	[87]
		MIRA-1	Maleimide	Michael acceptor binding to mutant p53	Experimental and/or preclinical	[88]
		STIMA-1	Styrylquinazoline	Michael acceptor binding to mutant p53	Experimental and/or preclinical	[89]
		3-Benzoylacrylic acid	Benzoylacrylate	Binds to mutant p53 by Michael addition	Experimental and/or preclinical	[90]
		KSS-9	Piperlongumine	Microtubule poison; redox; Michael acceptor binding to mutant p53	Experimental and/or preclinical	[91]
		PK11007	Sulfonylpyrimidine	Binds to mutant p53 by nucleophilic aromatic substitution	Experimental and/or preclinical	[92]
		ZMC1 ZMC2 ZMC3 ZN-1	Thiosemicarbazone	Zn^2+^ chelator	Experimental and/or preclinical	[48]
		COTI-2	Thiosemicarbazone	Zn^2+^ chelator	Phase I for gynecological tumors and head and neck cancer	[93]
		SLM P53-1	Tryptophanol-derived oxazoloisoindolinone	restores wt-like DNA binding ability to mut p53R280K Bridges extra interaction between p53 and DNA that rescues DNA binding and transcription activity	Experimental and/or preclinical	[94,95]
		SLM p53-2	Tryptopha-nol-derived oxa-zoloisoindolinone	Restores wild-type-like conformation and DNA-binding ability, possibly by enhancing interaction with Hsp70.	Experimental and/or preclinical	[96]
		MB725 MB710	Aminobenzothiazole	Binds to Y220C of p53 DBD	Experimental and/or preclinical	[46]
		PK083 Pk9318	Carbazole	Binds to Y220C of p53 DBD		[44,45]
		pCAPs	Peptides	Binds to mutant p53 and promotes refolding	Experimental and/or preclinical	[97]
	Mutant p53 degradation	Ganetespib Onalespib Luminespib TAS-116		Depletion of mutant p53 using HSP90 inhibitors or statins	In clinical trials	[9,98]
		Vorinostat	Suberanilohydroxamic acid (SAHA)	Histone deacetylase (HDAC) inhibitor, destabilizes mut p53 through inhibition of the HDAC6-HSP90 chaperone axis, and at the same time, inhibit the transcription of mutant p53 through HDAC8	In clinical trials	[72,77,99,100]
		Atorvastatin Lovastatin		Statin drugs, Inhibition of mevalonate pathway	In clinical trials	[9,101]
		NSC59984		Activation of MDM2		[21]
		Spaurtin		Chaperone-mediated autophagy (CMA) pathway		[102]
		Reacp53	Peptide	Disrupts mutant-p53 aggregates	Experimental and/or preclinical	[8]
	Interruption of mutant GOF	RETRA	2-(4,5-Dihydro-1,3-thiazol-2-ylthio)-1-(3,4-dihydroxyphenyl) ethanone	Binds to mutant p53 and disrupts mutant-p53–p73 complexes	Experimental and/or preclinical	[103]
		Prodigiosin		Disrupts mutant-p53–p73 complexes	Experimental and/or preclinical	[22]
	Immunotherapy	H2-scDb H2-Fab	Bispecific antibody	Bispecific antibody links T cells to cancer cells with one arm binding to T cell receptor and the other arm binding to HLA-mutant p53 R175H peptide on cancer cell surface.	Experimental and/or preclinical	[104]

Abbreviations: HSP40, heat shock protein 40; MDS, myelodysplastic syndrome; MQ, methylene quinuclidinone; p73, tumor suppressor protein p73 (p53 family member). Data partially reviewed from references [9,10,77,105,106].

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
