# Peer review of "Advanced Strategies for Therapeutic Targeting of Wild-Type and Mutant p53 in Cancer"

_biomolecules, 2022, doi:10.3390/biom12040548_

Round 1
Reviewer 1 Report
I would like to thank the authors for providing very interesting and comprehensive review entitled “Advanced strategies for therapeutic targeting of wild-type and mutant p53 in cancer” on the current achievements in targeting p53 in cancer treatment. The subject is of great scientific interest, and also very complex. The authors deal with the topic in a complete and detailed, very informative way and the main literature data are reported.
I have only few remarks:
Abstract – “P53 is the most frequently mutated gene”, should be TP53
Page 2, chapter 2 “Other cancer-associated mutations in TP53 are most often missense mutations in the central DNA-binding domain.” Since it is previously written about polymorphisms which are not considered mutations, I suggest to change the beginning of this sentence, e.g. “The” instead of “Other”
Page 4, Inconsistences in writing knockin and knockout/knock in and knock out/knock-in; P53 and p53…
The structure of p53 and several p53 mutants are nicely described. However, a figure and table related to sections 2.1. and 2.2. would contribute a lot.
The existence of cancer specific p53 isoforms with potential therapeutic potential should be at least mentioned.
Author Response
Point-by-Point Reply to Reviewer 1 Comments:
We thank Reviewer 1 for their constructive comments and careful review of our manuscript that we have significantly revised and improved. Below, we have a Point-by-Point response to the comments and concerns of reviewer 1.
Comments and Suggestions for Authors from the reviewer #1
I would like to thank the authors for providing very interesting and comprehensive review entitled “Advanced strategies for therapeutic targeting of wild-type and mutant p53 in cancer” on the current achievements in targeting p53 in cancer treatment. The subject is of great scientific interest, and also very complex. The authors deal with the topic in a complete and detailed, very informative way and the main literature data are reported.
I have only few remarks:
Abstract – “P53 is the most frequently mutated gene”, should be TP53
Response: Thank the reviewer for the suggestions. We have corrected gene “P53” to “TP53” in the current manuscript.
Page 2, chapter 2 “Other cancer-associated mutations in TP53 are most often missense mutations in the central DNA-binding domain.” Since it is previously written about polymorphisms which are not considered mutations, I suggest to change the beginning of this sentence, e.g. “The” instead of “Other”
Response: We have corrected it as the reviewer suggested.
Page 4, Inconsistences in writing knockin and knockout/knock in and knock out/knock-in; P53 and p53…
Response: We thank the reviewer for pointing it out. Knock-in or knockout has been written consistently throughout the manuscript.
The structure of p53 and several p53 mutants are nicely described. However, a figure and table related to sections 2.1. and 2.2. would contribute a lot.
Response: We thank the reviewer for the suggestions. We have made a figure to summarize p53 functional domains and utilization of p53 functional domain for drug development (Figure 1).
The existence of cancer specific p53 isoforms with potential therapeutic potential should be at least mentioned.
Response: We Thank the reviewer for the suggestions. We have made a brief description and discussion of p53 isoforms in the section 2.1 (page 5).
Reviewer 2 Report
The manuscript entitled: Advanced strategies for therapeutic targeting of wild-type and mutant p53 in cancer by Shengliang Zhang, Lindsey Carlsen, Liz Hernandez Borrero, Attila A. Seyhan, Xiaobing Tian and Wafik S. El-Deiry describes in detail increased of the knowledge about p53 in cancer.
Wafik S. El-Deiry and his collaborators presented a well-rounded review of the current advances in targeting wild-type and mutant p53 for cancer therapy. On the basis of what they have well described they consider the applicability of biotherapeutic and biopharmaceutical methods to increase the activity of p53 in cancer, providing new independent strategies dependent on p53 and p53 to aim at the functional restoration of the p53 pathway in p53 mutated cancer. The manuscript is well organized in the description of p53 structure. It is well articulated in the representation of the p21 target genes regulating the cell cycle regulating in turn the apoptosis and of the regulatory genes of cellular metabolism and intracellular immune functioning.
The research group studied some small molecule modulators of p53 family signaling and their antitumor effects demonstrating knowledge and experience about p53 and a possible use of these molecules in cancer therapy.
Figure 1 regarding the Strategies for targeting mutant p53 and wild-type p53 in cancer cells and Figure 2 regarding the Restoration of p53 pathway signaling in canonical and non-canonical regulatory pathways in cancer cells are explanatory and well represented as well as table 1 regarding the Overview of mutant-p53 targeting and wild-type p53 activating agents.
The authors conclude the manuscript indicating that although there are no FDA approved drugs targeting p53 (wild type or mutant), many studies and new strategies regarding the structure and function of p53 are considered and developed for application in the future in cancer therapy p53 related. Several small molecules have been considered by the authors for a targeted therapy, but to this day they indicate that improvements must be made to these therapeutic strategies. The contribution of the in-depth literature that the authors are familiar with is noteworthy.
Author Response
We thank reviewer 2 for their comments which are all positive regarding the review.
Reviewer 3 Report
The short review of Zhang et al is well written and gives a good resume about the most relevant strategies to target wild-type and mutant p53 in cancer. However, the innovation is rather limited as several more detailed reviews have been published recently on the same topic. In fact, some of them are missing in the reference section and should be cited in this review before publication. For example, the following ones should be cited in the manuscript:
- New therapeutic strategies to treat human cancers expressing mutant p53 proteins, JOURNAL OF EXPERIMENTAL & CLINICAL CANCER RESEARCH, 2018, Volume37, Article Number30, DOI10.1186/s13046-018-0705-7
- An Update on MDMX and Dual MDM2/X Inhibitors, CURRENT TOPICS IN MEDICINAL CHEMISTRY, 2018, Volume18, Issue8, Page647-660, DOI10.2174/1568026618666180604080119
- The multiple mechanisms that regulate p53 activity and cell fate, NATURE REVIEWS MOLECULAR CELL BIOLOGY, 2019, Volume20, Issue4, Page199-210, DOI10.1038/s41580-019-0110-x
- Small Molecules Targeting Mutant P53: A Promising Approach for Cancer Treatment, CURRENT MEDICINAL CHEMISTRY, 2019, Volume26, Issue41, Page7323-7336, DOI10.2174/0929867325666181116124308
- Current developments of targeting the p53 signaling pathway for cancer treatment, PHARMACOLOGY & THERAPEUTICS, 2021, Volume220, Article Number107720, DOI10.1016/j.pharmthera.2020.107720
- Recent Progress and Clinical Development of Inhibitors that Block MDM4/p53 Protein-Protein Interactions, JOURNAL OF MEDICINAL CHEMISTRY, 2021, Volume64, Issue15, Page10621-10640, DOI10.1021/acs.jmedchem.1c00940
Suggestions to improve the manuscript:
- Page 2, second paragraph, the authors should cite more updated reviews in the end of the sentence “….clinical trials.” and highlight more promising small molecule MDM2 inhibitors than the nutlins.
- In section 2.2. some other chemical scaffolds, reported more recently than PhiKan083 and SCH529074, should be mentioned. Moreover, Rita and SJ172550 should not be included in this section as there are reports that these are non-specific for p53, including mutant p53, and they are discussed latter in a more adequate place.
- In table 1, SLMP53-1 and other related derivatives with promising activities for mutant p53 are missing. Table 1, row MDM2/MDM4 inhibitors, it should be clear which compounds act as MDM2 inhibitors, which compounds act as MDMX inhibitors and which compounds act as dual inhibitors. COTI-2 should be mentioned in the discussion, not just in table 1.
- In figure 1, other small molecules should be included (especially the ones currently in clinical trials). MDM4 should be also included.
- the authors should add in the conclusion a brief overview of the problems associated to the use of MDM2 inhibitors and the interest of developing dual MDM2/4 inhibitors.
Minor details:
- Page 1, I suggest deleting in the text “(yet relatively unexploited)”.
- Page 2, first paragraph, correct “ansd”
- In some sections of the manuscript the discussion is very resumed. I suggest deleting subtitles 2.1.1 to 2.1.5 and keep the discussion of those subtitles in the same subtitle 2.1. Moreover, in section 2.1.4, some reviews about MDM2 inhibitors should be cited to support the discussion, and MDMX should also be mentioned in that subsection.
- Avoid the use of personal citation along the manuscript, as “Our lab has generated” and replace by third person citation or similar, as “A tool was developed to…”
- Section 3, page 6, cite relevant reviews in the end of the paragraph and include “Table 1” in the last sentence of the paragraph.
- Section 5.1, replace “MDM2 inhibitor ALRN-6924” by “MDM2/4 dual inhibitor ALRN-6924”
- Reference 22, the year is missing
Author Response
Point-by-Point Reply to Reviewer 3 Comments:
We thank Reviewer 3 for their constructive comments and careful review of our manuscript that we have significantly revised and improved. Below, we have a Point-by-Point response to the comments and concerns of reviewer 3.
Comments and Suggestions from Review #3
The short review of Zhang et al is well written and gives a good resume about the most relevant strategies to target wild-type and mutant p53 in cancer. However, the innovation is rather limited as several more detailed reviews have been published recently on the same topic. In fact, some of them are missing in the reference section and should be cited in this review before publication. For example, the following ones should be cited in the manuscript:
New therapeutic strategies to treat human cancers expressing mutant p53 proteins, JOURNAL OF EXPERIMENTAL & CLINICAL CANCER RESEARCH, 2018, Volume37, Article Number30, DOI10.1186/s13046-018-0705-7
An Update on MDMX and Dual MDM2/X Inhibitors, CURRENT TOPICS IN MEDICINAL CHEMISTRY, 2018, Volume18, Issue8, Page647-660, DOI10.2174/1568026618666180604080119
The multiple mechanisms that regulate p53 activity and cell fate, NATURE REVIEWS MOLECULAR CELL BIOLOGY, 2019, Volume20, Issue4, Page199-210, DOI10.1038/s41580-019-0110-x
Small Molecules Targeting Mutant P53: A Promising Approach for Cancer Treatment, CURRENT MEDICINAL CHEMISTRY, 2019, Volume26, Issue41, Page7323-7336, DOI10.2174/0929867325666181116124308
Current developments of targeting the p53 signaling pathway for cancer treatment, PHARMACOLOGY & THERAPEUTICS, 2021, Volume220, Article Number107720, DOI10.1016/j.pharmthera.2020.107720
Recent Progress and Clinical Development of Inhibitors that Block MDM4/p53 Protein-Protein Interactions, JOURNAL OF MEDICINAL CHEMISTRY, 2021, Volume64, Issue15, Page10621-10640, DOI10.1021/acs.jmedchem.1c00940
Response: We thank the reviewer for the suggestions. All the suggested references have been cited at their relevant parts in the current manuscript.
Suggestions to improve the manuscript:
Page 2, second paragraph, the authors should cite more updated reviews in the end of the sentence “….clinical trials.” and highlight more promising small molecule MDM2 inhibitors than the nutlins.
Response: We thank the reviewer for the suggestions. We have added more citations to support “the small molecule targeting mutant p53 under clinical evaluation”, and highlighted some of the promising MDM2 inhibitor AMG232 and dual inhibitor ALRN-6924 which are currently evaluated in clinical trials (page 2).
In section 2.2. some other chemical scaffolds, reported more recently than PhiKan083 and SCH529074, should be mentioned. Moreover, Rita and SJ172550 should not be included in this section as there are reports that these are non-specific for p53, including mutant p53, and they are discussed latter in a more adequate place.
Response: We thank the reviewer for the suggestions and comments. We have added more compounds such as PK9318, MB710 and MB725 which are new Y220 cavity binders in the section 2.2 (page 5)
RITA and SJ172550 have been removed from the last paragraph of the section2.2.
In table 1, SLMP53-1 and other related derivatives with promising activities for mutant p53 are missing. Table 1, row MDM2/MDM4 inhibitors, it should be clear which compounds act as MDM2 inhibitors, which compounds act as MDMX inhibitors and which compounds act as dual inhibitors. COTI-2 should be mentioned in the discussion, not just in table 1.
Response: We thank the reviewer for the suggestions. We have added SLMp53-1, SLMp53-2, MB710, MB725, PK083 and its analogous PK9318 in the raw of restoration of wild-type function to mutant p53 in table 1. We have generated two rows to separate dual inhibitors of MDM2/MDMX(MDM4) from the MDM2 inhibitors, and added two more small molecules as dual inhibitors of MDM2/MDMX(MDM4) in the table 1. We have made a brief description and discussion of COTI-2 in the section 4.1.
In figure 1, other small molecules should be included (especially the ones currently in clinical trials). MDM4 should be also included.
Response: We have revised the figure1 (which is renamed as figure 2 in the current version) as the reviewer suggested
the authors should add in the conclusion a brief overview of the problems associated to the use of MDM2 inhibitors and the interest of developing dual MDM2/4 inhibitors.
Response: We thank the reviewer for the suggestions. We have briefly stated the problems related to the administration of MDM2 inhibitors in the clinical studies in the summary and prospect.
Minor details:
Page 1, I suggest deleting in the text “(yet relatively unexploited)”.
Response: We have deleted them.
Page 2, first paragraph, correct “ansd”
Response: The typo has been corrected.
In some sections of the manuscript the discussion is very resumed. I suggest deleting subtitles 2.1.1 to 2.1.5 and keep the discussion of those subtitles in the same subtitle 2.1. Moreover, in section 2.1.4, some reviews about MDM2 inhibitors should be cited to support the discussion, and MDMX should also be mentioned in that subsection.
Response: We thank the reviewer for the suggestions. We have reorganized the section 2 by deleting the subtitles 1.2.2 to 2.1.5 sections and added more descriptions of MDM2/X with more citations. We also added one paragraph of brief review of p53 isoforms.
Avoid the use of personal citation along the manuscript, as “Our lab has generated” and replace by third person citation or similar, as “A tool was developed to…”
Response: We thank the reviewer for the suggestions. We have corrected these as the reviewer suggested.
Section 3, page 6, cite relevant reviews in the end of the paragraph and include “Table 1” in the last sentence of the paragraph.
Response: We thank the reviewer for the suggestions. Three relevant reviews and Table 1 have been added in the last sentence of the paragraph.
Section 5.1, replace “MDM2 inhibitor ALRN-6924” by “MDM2/4 dual inhibitor ALRN-6924”
Response: It has been corrected to MDM2/MDMX(MDM4) dual inhibitor.
Reference 22, the year is missing
Response: This reference has been removed from the current version of the manuscript.
Round 2
Reviewer 3 Report
The authors have addressed almost all points raised by this reviewer. However, it would improve the paper if all compounds included in table 1 are mentioned in the discussion of the manuscript.
Minor comments: In table 1, please correct to "SLM P53-1" to "SLMP53-1" and "SLM p53-2" to "SLMP53-2"